# Characterization of the Compounds Present in *Bougainvillea x buttiana* (var. Rose) with Healing Activity in a Murine Model

**DOI:** 10.3390/ph18050752

**Published:** 2025-05-19

**Authors:** Luís Martínez-Cuevas, Mayra Cedillo-Cortezano, Blanca Nury Echeverria Guerrero, Rodolfo Abarca-Vargas, Vera L. Petricevich

**Affiliations:** Facultad de Medicina, Universidad Autónoma del Estado de Morelos (UAEM), Calle Leñeros, Esquina Iztaccíhuatl s/n Col. Volcanes, Cuernavaca C.P. 62350, Morelos, Mexico; luis_123b@hotmail.com (L.M.-C.); mayra.cedillo@docentes.uaem.edu.mx (M.C.-C.); blanca.echeverriague@docentes.uaem.edu.mx (B.N.E.G.); rodolfo.abarca@uaem.mx (R.A.-V.)

**Keywords:** wound healing, acetonic extract, fractions, cytokines

## Abstract

**Background/Objective:** *Bougainvillea x buttiana* of the Nyctagenaceae family is widely used in traditional Mexican medicine for treating different diseases. This study was planned to estimate the healing effect of the acetonic extract obtained from *Bougainvillea x buttiana* (var. Rose). **Methods:** The bracts with flowers were subjected to extraction using maceration and concentrated in vacuo. Fractionation with a similar profile resulted in 11 fractions, which were determined using TLC. A mouse wound excision model was tested to evaluate the wound healing effect of the topical treatment pre-formulated with fractions of acetonic extract, which were determined using image analysis techniques. Cytokine levels present in the sera of mice treated or not treated with the acetonic extract were determined using the ELISA method. **Results:** The results obtained showed that the crude acetonic extract of *B. x buttiana* and/or its fractions in a pre-formulated hydrogel had wound healing capacity. The wound contraction rate and the healing speed in groups of animals treated with the pre-formulated crude extract and/or its fractions were significantly higher compared with the negative control (*p* < 0.001). Fraction 2 demonstrated more significant healing, reduced the production of cytokines such as IL-6 and TNF-α, and enhanced the levels of IL-10. **Conclusion:** The present study showed that the fractions obtained from the acetonic extract of *B. x buttiana* bracts were able to accelerate the wound healing process through anti-inflammatory mechanisms by regulating inflammatory cytokines. The results presented demonstrate that the extracts from *B. x buttiana* contain compounds that may be responsible for their healing properties.

## 1. Introduction

The skin is an organ that performs a wide variety of functions, including protective, thermoregulatory, sensitive, secretory, immunological, vitamin D-producing, and excretory functions [1]. The skin plays a role as a defensive barrier against physical damage and fluid loss and restricts the invasion of noxious substances [2,3]. The injury to the integrity of living tissue can be considered a wound, which is a natural physiological reaction to tissue injury, and it is a phenomenon that involves a complex process of continuous interactions between cells. The healing process can be divided into four phases: hemostasis, inflammation, proliferation or granulation, and remodeling or maturation [4].

The hemostasis phase prevents further bleeding immediately upon injury and is characterized by secreted substances that attract phagocytic cells, marking the start of the inflammatory phase. Immune cell recruitment is facilitated through chemokines, vasodilation, and increased blood vessel permeability. The first cells involved in the wound are the polymorphonuclear cells, or neutrophils, that are responsible for removing bacteria, viruses, or foreign particles from the wound surface. Hours after the injury, the appearance of neutrophils is characterized as a natural response of the body to fight inflammation and infection. Macrophages then appear after about 48 to 96 h to aid in recovery and replace neutrophils. One of the most important functions of macrophages is the removal of cellular debris, thereby defending against infections and promoting tissue regeneration [5,6,7].

The inflammation phase consists of an immune barrier against pathogens and is characterized by abundant neutrophil and monocyte infiltration and an increase in the levels of proteases and reactive oxygen species [8]. During the normal wound healing process, monocyte recruitment to the wound surface occurs with the depletion of neutrophils in response to chemotactic stimuli that are initiated due to both the migration of monocytes to the vascular wall and contact with the vascular endothelium, mediated by the participation of adhesion molecules of the selectin family, carbohydrate ligands, or integrins [9,10].

Monocyte movement is promoted by the secretion of cytokines and chemokines, like intercellular adhesion molecules (ICAMs) and vascular cell adhesion molecules (VCAMs), secreted by endothelial cells, to facilitate firm adherence to the endothelial wall. The migration of monocytes together with chemotactic gradients causes aggregation in the hypoxic, avascular surfaces of the wound.

Through its integrin and selectin receptors, binding occurs with specific proteins, such as CAMs, in the extracellular matrix. Within approximately 42 h of migration, monocytes are stimulated by bacterial products or diverse cytokines, such as IL-4, IL-10, IFN-γ, IL-13, as well as ECM components, to differentiate into macrophages. Monocytes that migrate to the wound commonly expand into an inflammatory or debridement phenotype, depending on the signals generated in the circulating tissue [11]. When the inflammatory response is prolonged or exacerbated, there is a delay in the subsequent phases of adequate wound healing and scar formation. There is ample evidence that the pro-inflammatory cytokines released by macrophages, such as IL-1β, IL-6 and TNF-α, are involved in the regulation of inflammatory reactions and the pain process, while wound healing is accelerated by the appropriate timing of the negative regulation of pro-inflammatory cytokine levels. The production of anti-inflammatory agents to suppress pro-inflammatory cytokines is therefore necessary to reduce this inflammatory phase [12,13].

Proliferation or granulation phase provides a barrier to infection, and matrix tissue in granulation formation is characterized by macrophages, fibroblasts, and blood vessels [14,15]. Macrophages are important in the removal of senescent cells and debris within the wound. These cells are involved in the removal of dead cells or tissue undergoing apoptosis. Surviving macrophages that have not undergone apoptosis remain around the wound bed and effectuate other functions and can stimulate collagen production, angiogenesis, and re-epithelization [16]. The release of growth factors and cytokines is fundamental for carrying out repair in the different phases of healing and maturation [17,18].

The remodeling or maturation phase transforms granulation tissue into scar tissue by degrading the type III collagen deposited on the tissue and replacing it with type I collagen. This phase can take up two years or can continue indefinitely [14,15]. The remodeling phase is primarily dependent on tissue breakdown and ECM production, and macrophages play a crucial role in both processes. Disarrangement is promptly monitored, and the production of the ECM is predominantly regulated through fibroblasts. In the late normal phase of wound healing, it is represented by a reduced number of wound macrophages, which makes ECM production less likely. These changes, along with wound contraction, result in a decrease in the size and volume of the wound. Under pathological conditions, the number and functions of macrophages can be modified, producing abnormal scars [19].

The treatment used for scars aims to improve the appearance and mobility of the scar. The degree of success mainly depends on the type and nature of the scar [20]. Despite the different existing surgical alternatives and the pharmacological treatment commonly used in wounds, ulcers, and various skin conditions caused in most cases by pathological states or because of some type of trauma, the use of plants with medicinal properties is still an option for patients. Treatment with herbs or medicinal plants is currently a fundamental resource for human health. The elements present in plants are the basis for the development of modern medicine. These elements have a synergistic effect; that is, they can all interact at the same time, so that some effects can complement or enhance others and/or neutralize their potential negative effects [21,22]. Medicinal plants occupy an important place, since for a long time, they have been the main therapeutic resource used for curing different illnesses [22]. In this sense, it has been scientifically demonstrated that the use of plant products in the treatment of scars has worked favorably, which can promote blood clotting, fight infection, and promote healing. The wound healing process is important for restoring both the interrupted anatomical continuity and the disturbed functional state of the skin. Various studies in literature show the improvement of this process using plant extracts and their isolated compounds in animal models. The exact mechanism implicated in wound healing is still unclear due to the existence of several parameters involved in the process [23,24]. Another important point is the extent to which the wound healing speed can be influenced by the substances contained in the medicine to be administered. The healing process will be faster with drugs that can improve healing by stimulating faster growth of new skin cells, reducing inflammation, and minimizing wound contamination [25,26,27]. The medicinal properties of these plant species depend on the bioactive phytochemical constituents within different types of chemical families, such as alkaloids, essential oils, flavonoids, tannins, terpenoids, saponins, and mainly phenolic compounds, which possibly contribute to their pharmacologic effects [28,29]. Additionally, plants are a main source of peptides, which are protein molecules that can exist naturally or be derived from peptide sequences with important biological activity [30]. Semi-solid pharmaceutical forms with 70–90% water content are known as hydrogels, which are easier to apply to wounds. They support tissue epithelium and granulation, which are important wound healing processes [31,32]. In that regard, *Bougainvillea x buttiana* is a hybrid species obtained from the cross between *B. glabra* and *B. peruviana*. These plants are widely distributed in Mexico, and the main uses in traditional medicine that are attributed to them include analgesic, anti-asthmatic, mucolytic, and anti-flu properties, as well as treatment against gastritis and gastroduodenal ulcers [33,34,35,36,37,38]. The present study evaluated the healing effect of the acetonic extract and the fractions obtained from *Bougainvillea x buttiana* (var. Rose).

## 2. Results

### 2.1. Preparation and Fractionation of Acetonic Extract of Bracts from Bougainvillea x buttiana

The extraction process is an important step in obtaining compounds with pharmacological activity from plant materials [37]. One of the most important steps in the extraction process is choosing the solvent, which, due to its polarity, plays a significant role in both the ability and effectiveness of pharmacological activities. *Bougainvillea x buttiana* bracts and flowers were subjected to an exhaustive maceration process using the solvent acetone with a weight yield of 3000 mg. The material obtained was subjected to a mobile phase with increasing polarity: DCM 100%, binary mixtures of DCM/methanol (*v*/*v*) (90:10%; 85:15%; 100%; 50:50%), MeOH 100%, water 100%, and DKM 100%. This resulted in 39 fractions, which were collected in vials and separated according to their elution order and the coloration they showed during elution in the column. Each of these fractions was subsequently analyzed using thin-layer chromatography (TLC) and regrouped into 11 major fractions identified as F-1 to F-11 from similar chromatographic patterns (Figure 1).

### 2.2. Yield of Fraction Groups from Bougainvillea x buttiana

Table 1 shows the dry weight and yield of each fraction. The percentage yields were calculated based on a total of 2392.3 mg of acetone extract (79.74%). The highest yield percentages were obtained for F-8 (8.56%) and F-5 (8.13%), while the lowest yield percentage was observed in F-1 (6.22%) (Table 1). These results are in accordance with other studies that show extraction yield [34,35].

### 2.3. Preliminary Phytochemical Study of Fractions Using Semi-Quantitative Analysis in Reversed-Phase Thin-Layer Chromatography (TLC)

Each of the 11 fraction groups, F-1–F-11, and crude extract were subjected to thin-layer chromatography (TLC) analysis in both normal and reversed phases to preliminarily characterize and confirm the metabolites present. The chromatographic plates were developed by exposure to different wavelengths of UV light and by treatment with selective colorimetric reagents. In the normal phase, a toluene–ethyl acetate mixture was used as the mobile phase, while in the reversed phase, methanol–water was used. For the detection of terpenoids, anisaldehyde–sulfuric acid developers were used to reveal triterpene presence. In the case of flavonoids, 4-hydroxybenzaldehyde was used as the developing agent under UV light. The analysis of F-2 revealed the presence of triterpenoids. In parallel, a normal-phase analysis with 4-hydroxybanzaldehyde development revealed a deep purple band with an R_f_ of 0.97, indicative of the presence of pentacyclic triterpenes. This type of coloration is common in oleanane and ursane compounds, which react with aromatic aldehydes to form conjugated chromophores after heating in an acidic medium (Figure 2).

The possible simultaneous presence of flavonoids and triterpenes in F-2 is not only consistent with previous phytochemical reports on the *Bougainvillea* genus [34,35,36,37] but could also explain the efficacy observed in the wound healing model. Flavonoids have been documented to exert beneficial effects in the initial inflammatory phase of wound healing due to their ability to scavenge free radicals and stabilize cell membranes, while triterpenes such as oleanolic acid act in later phases, promoting cell migration, angiogenesis, and re-epithelialization [37]. This synergistic effect could contribute to faster and more efficient recovery of injured tissue. The preliminary results show that F-2 comprises flavonoids and triterpenes. These compounds, acting synergistically or individually, could participate in modulating the inflammatory process and stimulating the tissue regeneration observed in the experimental model. In this context, there is a need to isolate and structurally characterize the compounds present in F-2 and other fractions obtained in this study through further studies using preparative chromatographic and spectroscopic techniques to confirm their identity and pharmacological activity. This step is essential to validate the therapeutic potential of *Bougainvillea x buttiana* and advance toward the development of phytopharmaceuticals based on active compounds.

To determine the other families of compounds present in each fraction, each sample was subjected to other systems, such as an eluent system composed of distilled water–acetonitrile (10:90 or (70:30) and reversed-phase plates were used. The bands were developed using a specific reagent for flavonoids. The Rf value and areas observed for each fraction were compared with those of the rutin used as a standard. The results obtained using reversed-phase thin-layer chromatography (TLC) indicated the presence of flavonoids, based on the reddish fluorescence observed under long-wave UV light and the R_f_ value, which intensified after development with specific chromogenic reagents such as NP/PEG or aluminum chloride in ethanol. This type of response is characteristic of glycosylated or methoxylated flavones and flavonols, metabolites widely distributed in the plant kingdom (Figure 3).

In the systems used, it was possible to verify that for the water–acetonitrile (10:90) system, F-2 showed an Rf value and an area like those of rutin. For the water–acetonitrile (70:30) system, F-6, F-7, F-8, and F-10 had Rf values similar to those of rutin. Smaller areas were detected for F-6 and F-7 and larger areas for F-8 and F-10 compared with the areas for rutin (Table 2).

### 2.4. Topical Evaluation of the Acetonic Extract Obtained from Bougainvillea x buttiana

Experimental groups were formed as described above. The kinetics of wound closure and the macroscopic physical characteristics were evaluated, and the percentage of contraction was determined. The macroscopic study is shown in Figure 4. Different groups of animals were treated with KitosCell^®^, hydrogel (vehicle), or hydrogel + crude AEBxbR extract and/or fractions F-1, F-2, F-3, F-4, F-5, F-6, F-7, F-8, F-9, F-10, or F-11. In general, when carrying out an analysis of wound healing, the wound size progressively decreased throughout the 14 days.

Figure 5 shows a comparison of wound closure using different treatments. The wound size reduced in a time-dependent manner. A comparison of wound areas between the untreated group (negative control) and the group treated with hydrogel supplemented with crude extract AEBxbR is shown in Figure 5A. In the treatment with hydrogel + crude extract, the wound sizes decreased over time and were significantly smaller compared to those in the negative control (*p* < 0.001). A comparison of the wound size between treatment with hydrogel alone and hydrogel + crude extract AEBxbR is shown in Figure 5B. For the treatment with crude extract AEBxbR, the wound areas were significantly smaller compared to those obtained in treatment with hydrogel alone (*p* < 0.005). Finally, Figure 5C shows a comparison of the wound size between hydrogel + crude extract AEBxbR treatment and KitosCell^®^ treatment. The wound sizes from treatment with crude extract AEBxbR were significantly smaller compared with treatment with KitosCell^®^ (*p* < 0.05). The wound sizes in the first 5 days were similar for all treatments; the main differences in wound sizes were observed between the 6th and the 10th day in the groups treated with crude AEBxbR.

### 2.5. Comparison of Topical Treatment of Hydrogel + Crude Extract with Each of the 11 Fractions

The wound contraction percentages were compared between treatments with crude AEBxbR, its fractions, and KitosCell^®^. As shown in Figure 6, for treatments with hydrogel + crude AEBxbR and KitosCell^®^, the wound contraction percentage increased in a time-dependent manner, reaching a maximum of 95.45% and 95.34%, respectively, on the 14th day. Up to the 10th day of treatment, the wound contraction percentages were similar among all treatments. On the 11th day, the wound contraction percentages for the F-1, F-2, and F-7 treatments were slightly higher compared to those obtained for other treatments, reaching a maximum of 94.36%, 95.38%, and 93.88%, respectively (Figure 6). On the 14th day, the wound contraction percentages were 96.30% (F-1), 98.56% (F-2), and 97.75% (F-7) (Figure 6). From the 11th to the 14th day, the wound contraction percentages obtained from the treatments with F-3 (93.49% to 98.82%), F-4 (87.92% to 97.85%), F-5 (87.87% to 98.76%), F-6 (90.17% to 96.80%), F-8 (91.39% to 97.47%), and F-10 (82.45% to 94.58%) are shown in Figure 6. At the same time, the wound contraction percentages obtained for F-9 and F-11 treatments were 91.20% to 93.73% and 75.35% to 92.01%, respectively (Figure 6). For the KitosCell^®^ treatment, from day 11 to 14, the wound contraction percentages were 91.59% to 95.34%. The wound contraction percentage for F-2 treatment on day 11 was slightly higher compared to other treatments. On subsequent days, the wound contraction percentages continued to be higher for the F-2 treatment (Figure 6).

### 2.6. Determination of Healing Speed

The speed of healing depends on different factors that directly influence the size and location of the wound, hygiene, and the quality of the immune response [38,39]. Figure 7 shows the speed of healing for each of the experimental groups. The measurement of the wound was recorded from the beginning of the test until complete healing, and based on the results obtained, it was possible to determine the healing speed of each group at different times. The groups of animals treated with KitosCell^®^ showed a high speed of healing on the 1st day, with the speed gradually decreasing until the 14th day. On the contrary, in the groups of animals treated with AEBxbR, the healing speed increased until the 4th day and was maintained until the 14th day (Figure 7A). The F-1, F-3, and F-8 treatments showed identical healing speeds compared with AEBxbR (Figure 7 B,D,I). The F-4, F-5, F-7, F-9, and F-11 treatments showed higher healing speeds on the 1st day, which declined afterwards, and exhibited similar healing speeds compared with KitosCell^®^ treatment (Figure 7E,F,H,J,L). With respect to the 1st day of treatment with F-6 and F-10, the healing speeds were significantly higher compared to those obtained from treatment with KitosCell^®^ (*p* < 0.001) (Figure 7G,K). Figure 7C presents the F-2 treatment. The healing speed increased up to the 5th day and was significantly higher compared to treatments with AEBxbR and/or KitosCell^®^ (*p* > 0.001). From the 5th to the 9th day, a slight decrease in the healing speed was observed. Finally, from the 10th to the 14th day, the healing speeds were similar to those obtained with treatments with AEBxbR and/or KitosCell^®^ (Figure 7C). The healing speed varies depending on several factors, including the wound type and location and the age and health of the patient. To speed up healing, it is crucial to keep the wound clean and covered, applying heat and elevating the affected area if necessary. The faster healing speed obtained with the F-2 treatment suggests that this fraction can help the healing to progress more quickly.

Considering that the treatment with F-2 showed one of the highest percentages of wound contraction and the highest healing speed, we decided to use this treatment for the remaining analyses.

### 2.7. Histological Evaluation of F-2 Treatment

A histological evaluation was carried out for the negative control, positive control (KitosCell^®^), and groups of mice treated with F-2. The histological examination is shown in Figure 8, where healing changes were captured at 10X magnification and the corresponding indications labeled. For the negative control, on day 2, minimal polymorphonuclear (neutrophil) inflammatory infiltrate (Figure 8A) was observed. On day 6, the layer of granulation tissue (neutrophils) was observed to become dry and ulcerated (Figure 8D). On day 9 (Figure 8G), the size of the wound decreased and granulation tissue was observed up to the muscular layer. Then, mononuclear (lymphocyte) infiltration began (chronic inflammation). By day 12 (Figure 8J), the wound was still very superficial, the area of granulation (lymphocytes) continued, and the observation of fibrosis in the reticular dermis began. On day 14 (Figure 8M), complete re-epithelialization with retraction of the epithelium and fibrosis was observed.

For the positive control (KitosCell ^®^), on day 2 (Figure 8B), necrosis of the ulcer edge was visible along with the inflammatory infiltrate of polymorphonuclear cells (neutrophils). On day 6 (Figure 8E), ulceration continued to be observed up to the granulation tissue (neutrophils), and some areas with calcium deposits (microcalcifications) were identified. On day 9 (Figure 8H), the presence of mononuclear cells (lymphocytes) was observed in greater quantities than in the negative control, the size of the scab had decreased, and granulation tissue was observable up to the muscular layer. On day 12 (Figure 8K), the scab was observed to be very superficial and smaller than in the control, with this loss being secondary to the presence of re-epithelialization. On day 14 (Figure 8N), there was no retraction of the epithelium and less fibrosis than in the negative control.

For the F-2 treatment, on day 2 (Figure 8C), ulcer formation was almost complete; scarce necrosis, polymorphonuclear infiltrate (neutrophils), and formation of granulation tissue was seen. On day 6 (Figure 8F), the number of neo-formed vessels increased, extending to the muscular layer, and the presence of inflammatory infiltrate of mononuclear cells (lymphocytes) was observed (chronic inflammation). On day 9 (Figure 8I), almost total loss of the scab and the presence of collagen bands in the papillary dermis were observed, and the reticular dermis showed minimal fibrosis (scarring). On day 12 (Figure 8L), the presence of the scab was minimal and re-epithelialization was almost total; the infiltrate of mononuclear cells (lymphocytes) (chronic inflammation) as well as granulation tissue had decreased. On day 14 (Figure 8O), there was no retraction of the epithelium, and the more superficial fibrosis was similar to that observed in the positive control. In the positive control, the level of the epithelium is continuous with the formation of scar, while in the fraction, it is observed to be slightly elevated on day 14. However, until day 12, no overgrowth of the scar is observed; that is, the changes observed are perfect for healing, as also seen in Figure 8. The positive control shows a decrease in marks, which represents the difference observed on day 14 compared to the use of the fraction.

The analysis of the changes observed in the healing process reveals histopathological changes, such as changes in inflammatory infiltrate, granulation tissue, and finally fibrosis, which are described in Table 3. Through semi-quantitative analysis, a score is assigned based on the absence or presence of changes observed in the different parameters, characteristics, and days of treatment. It is worth noting that treatment with F-2 between days 2 and 6 presented lower scores for inflammatory infiltrate, granulation, and ulceration and greater angiogenesis compared to the controls. On day 9, treatment with F-2 showed less chronic inflammation, no scabs or fibrosis, and the beginning of collagen bands. On day 12, treatment with F-2 showed minimal scabs and fibrosis, and almost complete re-epithelialization. By day 14, there was no epithelial retraction or superficial fibrosis, and complete re-epithelialization. All parameters analyzed suggest that treatment with F-2 presented slightly better scores compared with the control treatments (negative and positive).

### 2.8. Effect of F-2 Treatment on Cytokine Production and Equilibration of Pro-Inflammatory and Anti-Inflammatory Cytokines

The healing process consists of different phases: the homeostasis phase, comprising a period 1 to 2 days; the inflammatory phase, from days 2 to 6; the proliferation or granulation phase, from days 6 to 9; and the maturation phase from days 9 to 12. The effect of treatment with F-2 on cytokine production was analyzed by comparing these phases (Figure 9). In the negative control without treatment, the production of pro-inflammatory cytokines increases in a time-dependent manner, attaining a maximum level on the 6th day—which coincides with the inflammation phase—and decaying thereafter. Similar behavior was observed with the F-2 treatment, although the levels of IL-6, TNF-α, and IFN-γ were significantly lower compared to the negative control (*p* < 0.001) (Figure 9). In both treatments, the maximum production of IL-4 and IL-5 was obtained on the 6th day and decayed thereafter. For the F-2-treated groups, the production of these cytokines was significantly lower compared with the levels obtained in the negative controls (*p* < 0.001) (Figure 9). With the F-2 treatment, the production of IL-10 increased up to the 6th day and remained high until the 12th day, decaying thereafter, coinciding with the final part of the inflammation phase and the proliferation and maturation phases. IL-10 levels were significantly higher in the groups treated with F-2 compared to the levels obtained in the control groups (*p* < 0.001) (Figure 9). Figure 9 also shows the results of the predominance of responses by calculating the ratio of pro-inflammatory/anti-inflammatory cytokines, where a value > 1 corresponds to a pro-inflammatory response and a value < 1 corresponds to an anti-inflammatory response. In the negative control treatment, for the IL-6/IL-10, TNF-α/IL-10 and IFN-γ/IL-10 balances, the predominant response was clearly pro-inflammatory on the 1st day and decreased until the 6th day (Figure 9). For the F-2 treatment, the predominant response was anti-inflammatory from the 1st to the 14th day (Figure 9).

## 3. Discussion

The present study showed that the acetone extract of *Bougainvillea x buttiana* was able to increase the healing rate of wounds caused by incisions. After the first seven days of wound infliction, the wound healing rate in the groups treated with F-2 was significantly higher compared to that found in the negative control group, with behavior identical to that of the positive control group (KitosCell^®^). This observation may be due to the fact that in the case of wounds caused by incisions, where there is minimal cell loss and no contamination, the tissue injury can be repaired in 4 days [1]. To preserve the physiological condition, in clinical practice, solutions for free surfactants that supposedly cause skin lesions are used for wound irrigation [20]. Furthermore, it is well documented that collagen is the dominant protein constituent of wound connective tissue that is considered responsible for tissue strength [21]. Thus, an increase in the healing activity of *Bougainvillea x buttiana* extract and its fractions may indicate an increase in collagen in the wound lesion. These findings corroborate previous studies that demonstrated an increase in collagen synthesis in wounds treated with extracts [11,12,22,23,24,25]. As described in the literature, there is significant cell and tissue destruction in burn wounds that significantly hinders the healing process [26]. Treatments with acetone extract from *B. x buttiana* and its fractions, with the exception of F-9 and F-11, showed improvements in wound contraction that were slightly higher compared to the positive treatment. Our results showed that on day 14, an analogous grade of wound healing was observed in all the groups treated with the extract and its fractions. The wounds treated with the extract seemed to show better healing compared to the negative controls.

The results obtained regarding the behavior of each of the experimental groups treated with AEBxbR and/or its fractions agree with those regarding the cellular components that act in each of the healing phases. In the first period, which corresponds to the hemostasis phase, fibrin, platelets, leukocytes, and macrophages are involved [17,18]. The reduction in the size of the wound in groups F-9, F-10, and F-11 is possibly due to the fact that the components present in these groups of fractions help to promote faster healing compared to the other treatments, reducing the initial size of the wound. The inflammatory phase, which occurs on days 4, 5, 6, and 7, is mainly characterized by the presence of macrophages, monocytes, and lymphocytes for cleaning and preventing infection in the wound [17,18]. In the period corresponding to the 8th and 9th days, different processes occur in which fibroblasts and capillaries intervene through the stimulation of endothelial cells without showing changes in the size of wounds [17,18]. Between the 10th and 11th days, collagen production, epithelial cells, and fibrous connective tissue intervene to replace the fibrin clot [17,18]. There is also a significant reduction in the size of the wound caused by each of the experimental treatments. In the last few days, the processes of epithelialization and wound remodeling occur in each of the experimental groups, with the active participation mainly of fibroblasts [17,18]. Our results suggest that components such as terpenes and flavonoids, present in each of the groups of fractions, are capable of actively promoting the healing process. Among the groups, F-8, F-9, and F-10 showed a high capacity to promote wound closure. In contrast, F-1, F-2, and F-3 did not show significant activity in wound closure during the initial stages of healing, and only after the inflammation phase, with the loss of the scab, was a smaller wound area observed. In general, analyzing the decrease in the size of the wound over the 13 days of the test, it was possible to verify that the groups F-1, F-2, and F-3 presented the highest percentage of wound contraction, at 96.30%, 98.56%, and 98.82%, respectively. Our results suggest that the behavior of each of the experimental groups agrees with the interaction generated by the applied treatment and the cellular components that act in each of the healing phases. The decrease in the size of the wounds observed in the experimental groups treated with F-9, F-10, and F-11 is possibly due to the components present in these groups of fractions, which help promote faster healing compared to the other groups. It is worth mentioning the presence of clean wounds, demonstrating the antibacterial activity of the species *Bougainvillea* [37,38,39].

The treatment with F-2 showed one of the highest wound contraction percentages and the highest healing speed; this treatment was used for the remaining analyses. The semi-quantitative analyses for wound re-epithelialization suggested that the treatment with F-2 presented slightly better scores compared with both the negative and positive controls. According to the results obtained regarding the production of pro-inflammatory cytokines with F-2 treatment, there was a significant decrease in relation to the negative control, reaching a maximum peak on the 6th day, corresponding to the inflammatory phase. These results indicate a similar expression pattern observed in terms of the production of pro-inflammatory cytokines, whereby treatment with F-2 stimulated a reduction in the pro-inflammatory cytokine expression, such as IL-6 and TNF-α. However, due to the nature of the wound, which can be classified as clean, there is evidence that the levels of this type of cytokine increase in more aggressive injuries or surgeries [37].

IL-6 is a glycoprotein secreted by macrophages, monocytes, and eosinophils, responsible for neutrophil maturation and activation, macrophage maturation, and T lymphocyte differentiation [38]. It is a cytokine that participates in inflammatory processes, and its secretion can be generated by different stimuli from LPS, TNF-α and IL-1β. In turn, it can exert a double effect, since the initial pro-inflammatory response is controlled by immunoregulatory molecules, such as specific inhibitors and soluble cytokine receptors [39]. During the injury process, plasmatic concentrations normally reach a peak in 4–6 h and may persist for up to 10 days [38,39,40].

TNF-α, which is produced by macrophages and helps to promote edema in harmful processes, acts as a vasoactive mediator [41]. Its decrease has been shown to block the inflammation phase and prevent oxidative stress and, therefore, fibrosis [41]. This anti-fibrotic effect modulates collagenase activity and the synthesis of glycosaminoglycan and increases the levels of glucocorticoids, cortisol, prolactin, and catecholamine that promote the reduction of levels of pro-inflammatory cytokines [42,43]. Additionally, studies have been found on the kinetics of expression of the plasmatic levels of pro-inflammatory molecules that mention an increase in the early phases of healing and an increase in anti-inflammatory molecules in later phases [44]. During the healing phase, from days 2 to 6, corresponding to the hemostasis and inflammation phases, an increase is observed in the levels of pro-inflammatory cytokines IL-6, TNF-α, and IFN-γ, which protect and remodel damaged tissue and develop better wound closure [45]. Similarly, there are cytokines that can inhibit the production of pro-inflammatory cytokines such as IL-10 [46]. This cytokine is secreted by monocytes and TH2 lymphocytes and is responsible for regulating the inflammatory response. In the literature, some studies have shown that in the presence of wounds, high levels of IL-10 develop for up to the first 10 days [36]. The F-2 treatment was able to increase IL-10 levels. In this study, in the groups of animals treated with F-2, the maximum expression was reached from days 8 to day 12, corresponding to the proliferation and maturation phases. The F-2 treatment was also able to stimulate the production of cytokines such as IL-4 and IL-5, which reached maximum expression in the inflammatory phase, on the 6th day. These cytokines are synthesized from TH2 cells and are characterized by their involvement in the development, modulation, and regulation of inflammation [47].

In our study, we observed that treatment with F-2 significantly accelerated wound closure during the inflammation phase. This highlights the anti-inflammatory effect of F-2 on the healing process, as demonstrated by photomicrographs capturing the progression of wound closure. The therapeutic potential of F-2 aligns with a broader understanding that natural products can enhance wound healing. Several studies have highlighted the ability of natural compounds to promote wound healing, particularly those with anti-inflammatory, antioxidant, and antibacterial properties, as well as their ability to stimulate collagen synthesis. Considering these positive effects, they are commonly associated with the presence of bioactive phytochemicals from diverse chemical groups, such as flavonoids, terpenes, tannins, saponins, and phytosterols [37]. Our preliminary results show the presence of these compounds, although further studies should be carried out to identify the mechanism of action of treatment with F-2, or even other fractions.

## 4. Materials and Methods

### 4.1. Reagents

KitosCell^®^ Gel PFD, propylene glycol, triethanolamine, silica gel, dichloromethane, and methanol were obtained from Sigma Aldrich Chemical Co. (Toluca, EM., Mexico). Capture and detection antibodies and recombinant cytokines were purchased from BD Biosciences Pharmingen (San Diego, CA, USA).

### 4.2. Collection and Identification of Vegetal Material

The collection of bracts with flowers from *Bougainvillea* species was carried out in the municipality of Temixco, Morelos, Mexico, (18°52′20.1″ N and 99°14′40.6″ W, 1.185 msm) in March 2018. For identification, one specimen was deposited in the HUMO Herbarium of the (CIByC) Biodiversity and Conservation Research Center (UAEM, Cuernavaca, Morelos, Mexico) and designated with folio number 33872 as *Bougainvillea x buttiana* Holtum and Standl (var. Rose).

### 4.3. Preparation of the Acetonic Extract of Bougainvillea x buttiana (AEBxbR)

Bracts with flowers were subjected to a drying process in the shade at a temperature of 23 ± 1.2 °C and a relative humidity of 25.7 ± 1.5%. Later, the dry plant material was ground, obtaining a particle size of 1.2 ± 0.2 mm. Then, it was subjected to an exhaustive maceration process using 100% acetone as solvent for a period of 72 h. Thereafter, the plant material was filtered and solvent removal was performed under reduced pressure using Rotavapor^®^ (Buchi R-100) (Heidolph, Elk Grove Village, IL, USA).

### 4.4. Primary Fractionation Using Open-Column Chromatography of Acetonic Extract

The acetonic extract of *B. x buttiana* (var. Rose) was subjected to fractionation using open-column chromatography. Here, 40 g of 60–200-mesh silica gel was used, and the elution was carried out using a mobile phase in increasing order of polarity, using the solvents dichloromethane (DCM) and methanol (MeOH), and subsequently washing with 100% methanol, water, and acetone. The fractions obtained during elution from the column were identified using thin-layer chromatography (TLC) in a dichloromethane–methanol (90:10, *v*/*v*) system and developed with iodine. The fractions with similarities in their chromatographic profile were grouped according to the spots observed in the TLC. Once grouped, the weight and yield of each of the fractions were calculated. Subsequently, a chromatographic analysis was carried out for the groups of fractions obtained using different developers in order to identify the components present in the Centro de Investigación Biomédica del Sur (CIBIS-IMSS). According to the physicochemical characteristics of each fraction, groups F-1 to F-4 were analyzed, as well as the crude acetone extract on a normal phase plate with an ethyl acetate–hexane (30:70) system with terpene and flavonoid standards. For the development of the plates, 4-hydroxybenzaldehyde was used and observed in UV at 254 nm and 365 nm. Groups F-5 to F-11 as well as the crude acetone extract were run on a reverse-phase plate with a water and acetonitrile system (70:30) using a routine standard, revealed with a flavonoid standard, and observed in UV at 254 nm and at 365 nm.

### 4.5. Topical Evaluation of Healing Activity: Animals and Treatments

#### 4.5.1. Animals

Female mice of the BALB/c strain with a weight between 25 and 30 g were acquired from the Bioterio of the Instituto Nacional de Salud Pública, Cuernavaca, Morelos and manipulated based on the Federal Regulation for the Management and Experimentation of Animals, issued by the Secretaría de Agricultura y Recursos Naturales (SAGARPA) and in accordance with the procedures described in NOM-062-ZOO-1999 [48]. The animals were intended solely for experimental use and were kept in the laboratory for a two-week acclimatization period. They were grouped in boxes of six animals and kept under controlled temperature conditions at 25 ± 1 °C, with a diet of commercial rodent pellets and water ad libitum. The experiments designed for this study were approved by the Committee of Experimental Animal Administration of the university with protocol number 11MMM2019.

#### 4.5.2. Surgical Procedure for the Realization of Skin Wounds

Groups of female mice were anesthetized with isoflurane, which is a strong anesthetic, via inhalation using a gas chamber (20 µL for induction) [49]. Subsequently, the upper dorsal area was shaved and sterilization was performed with alcohol swabs 10 mm in diameter and 4–5 mm thick. An incision was made with the help of a 4.0 mm uni-punch skin biopsy needle. After the incision was made and excess blood was cleaned with an alcohol swab [50]. The wounds were kept open, and the possibility of the existence of some type of secondary infection was ruled out using macroscopic analysis. Topical treatment was applied every 12 h during the first three days using pre-formulated hydrogel as a vehicle. The pre-formulated hydrogel was prepared as described in Table 4.

#### 4.5.3. Treatment with Hydrogel Application

The hydrogel, prepared as described above, was applied topically to the dorsal wound of each mouse using a sterile micropipette in a volume of 20 µL per day. Administration took place during the first three days after wound induction, i.e., before scab formation. This strategy is based on previous evidence indicating that the first days of the healing process are critical as they correspond to the early inflammatory phase and the beginning of the proliferative phase, moments in which topical treatments can most effectively modulate the tissue response [51]. The hydrogel was applied directly to the wound, without the need for a dressing or bandage, which is common in rodent wound healing models, where the size of the lesion, the low physical activity of the animal under controlled conditions, and the viscosity of the hydrogel favors its permanence at the application site [52]. In this sense, no detachment of the gel was observed, nor did it interfere with the evolution of the healing process or with the parameters evaluated.

During the healing phase, the size of the wound was measured on days 0, 2, 6, 11, and 13 of the trial using a Canon T6 camera with an 18–55 mm lens. Photos of the wounds were taken throughout the trial, and the healing area was determined using the Image J-1.53t (https://imagej.net/ij/ Java 1.8.0_345 (ImageJ is in the public domain) (accessed on 1 February 2025)). To determine the wound healing effect of AEBxbR, the groups of fractions were organized as described in Table 5.

#### 4.5.4. Macroscopic Assessment of Wound Contraction

A macroscopic evaluation was carried out using the average values of the areas of the wounds obtained, in which the degree of wound contraction was calculated and expressed as a percentage using the following formula [53]:Wound contraction percentage (%) = [(A_0_ − A_F_)/A_0_] × 100
where A_0_ is the initial area, corresponding to the day of the surgical procedure, and A_F_ is the final area on specific days.

During the macroscopic analysis and the evolution of wound closure in each of the experimental groups, the characteristics of the wound, such as the presence of exudate, edema, and the occurrence of hemorrhage, were evaluated and analyzed.

#### 4.5.5. Determination of Healing Speed

During the test, the healing speed per day in each experimental group was determined using the following formula:Healing speed = Initial area − Area at time x/Hour X

### 4.6. Histopathological Study with Hematoxylin–Eosin (HE) Staining

For the histopathological study, the animals were divided into three groups: those who received treatment with F-2, a negative control, and a positive control (KitosCell^®^). Previously, the mice were anesthetized using a gas chamber with 20 μL of isoflourane, and skin samples were taken by making cuts around the wounds on days 0, 2, 6, 9, 12, and 14, corresponding to the healing phases. Subsequently, the skin samples were placed in a 10% formalin solution. After exposure, the tissues were washed with running water and dehydrated in an ascending mixture of ethyl alcohol and xylene. Then, 3 µm paraffin-embedded tissue sections were cut and mounted on glass slides. The histological sections were stained using the H&E technique for histological examination. Digital photomicrographs were taken for each sample with a digital camera attached to a microscope. The tissue samples were evaluated for the presence of hemorrhages, epidermal exfoliation, crust formation, inflammatory cellular responses, and fibrous tissue proliferation.

### 4.7. Determination of Cytokines Using the ELISA Method

To compare the effect of treatment with the negative control and F-2 on cytokine production, two groups of animals were randomly separated, and blood from these animals was collected. The cytokines present in the sera from each treatment were examined in detail, according to the method described by Arteaga et al. (2017) [33]. The evaluation of cytokines was performed in accordance with the manufacturer’s instructions. All results were expressed in pg/mL; the minimum detection values were 10 pg/mL for cytokines IL-4, IL-5, IL-6, and TNF-α and 100 pg/mL for IFN-γ.

### 4.8. Statistical Analysis

The results obtained were presented as mean ± standard deviation (n = 5) and a statistical analysis of variance (ANOVA) was conducted, followed by a Tukey test for independent samples with the help of the GraphPadPrism 8 program. *p*-Values < 0.05) were considered significant.

## 5. Conclusions

The results obtained in this study showed that the acetonic extract of *Bougainvillea x buttiana* has a healing effect. Of all fractions obtained, F-2 was characterized by the presence of flavonoids and terpenes and exhibited both a higher rate of healing compared to the other experimental groups, reaching a peak on the 8th day, and an increment in the levels of IL-10.

## Figures and Tables

**Figure 1 pharmaceuticals-18-00752-f001:**
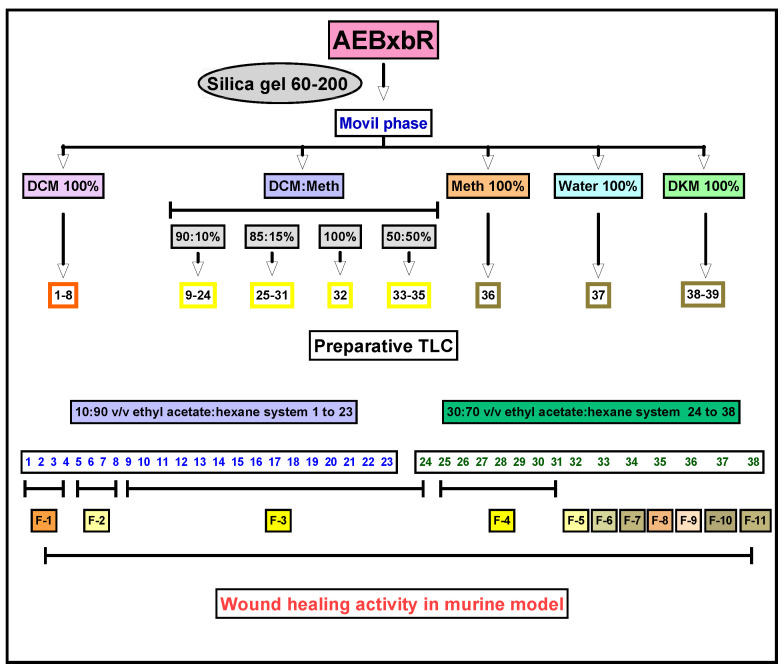
Experimental strategy for the bioassay-guided fractionation of wound healing activity.

**Figure 2 pharmaceuticals-18-00752-f002:**
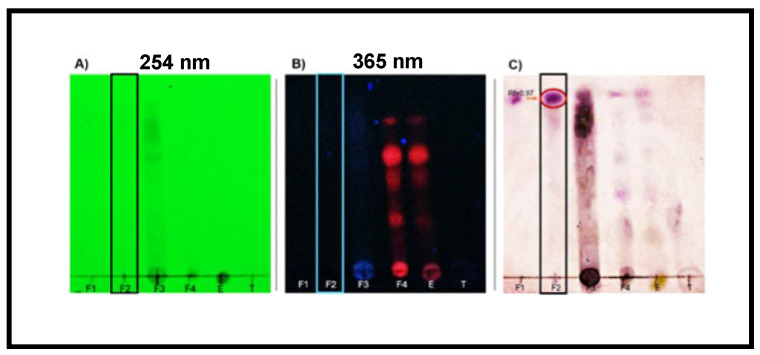
Thin-layer chromatography (TLC). Normal chromatography of fraction groups 1 to 4 at different UV wavelengths. TLC plates developed on silica gel and visualized under (**A**) short-wave UV light (254 nm) and (**B**) long-wave UV light (365 nm). (**C**) Terpene developer revealed a distinct purple band at R_f_ ≈ 0.9, indicative of pentacyclic triterpenoids such as oleanolic acid or related compounds.

**Figure 3 pharmaceuticals-18-00752-f003:**
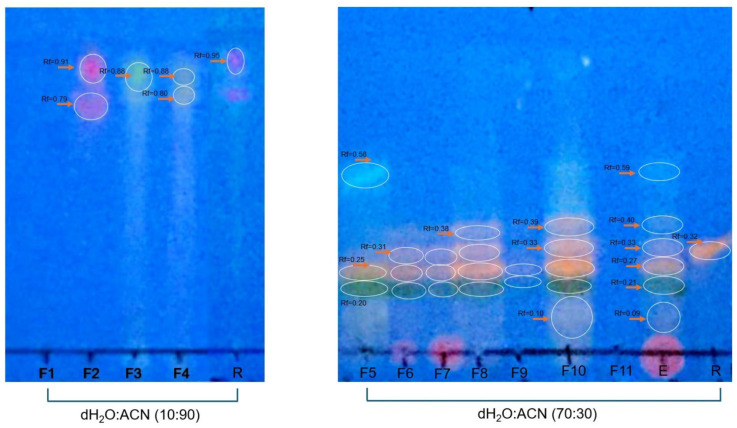
Analysis of flavonoid presence. Normal TLC for F-1, F-2, F-3, and F-4 developed in a water–acetonitrile (10:90, *v*/*v*) system. Reversed-phase thin-layer chromatography (RP-TLC using RP-18 F254s plates) for F-5 to F-11 developed in a water–acetonitrile (70:30 *v*/*v*) system. The plates were developed with the 4-hydroxybenzealdehyde reagent and evaluated under UV light at 365 nm. Fraction groups 1 to 11 were compared with rutin (1 µg/mL) used as standard.

**Figure 4 pharmaceuticals-18-00752-f004:**
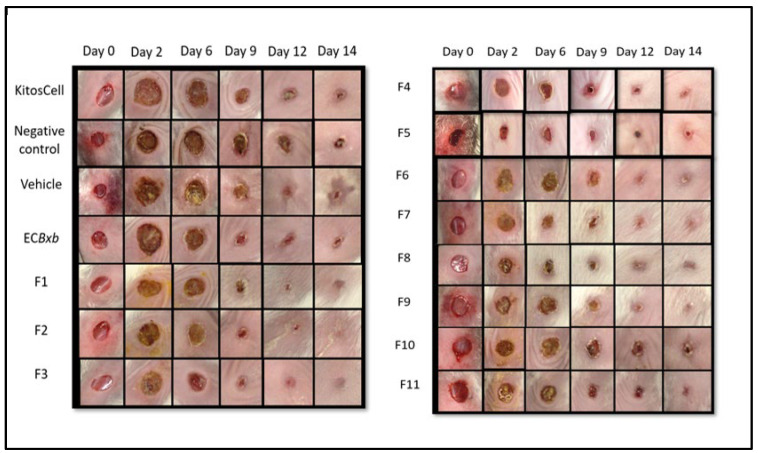
Macroscopic comparison of wound closure between the different experimental groups on days 0, 2, 6, 9, 12, and 14.

**Figure 5 pharmaceuticals-18-00752-f005:**
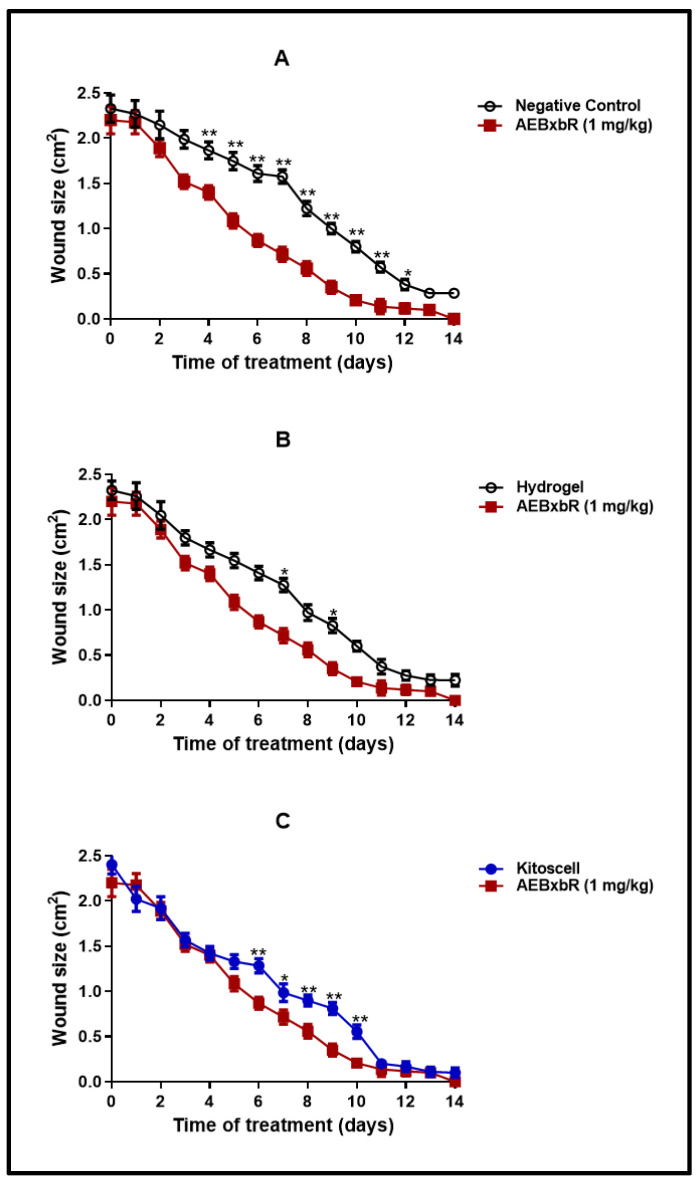
Wound size from days 0 to 14. Values correspond to the mean (*n* = 6, +SEM) of the two-way ANOVA statistical analysis using Tukey’s post hoc test (* *p* < 0.001, ** *p* < 0.005). (**A**) A comparison of wound areas between the untreated group (negative control) and the group treated with hydrogel supplemented with crude extract AEBxbR. (**B**) A comparison of the wound size between treatment with hydrogel alone and hydrogel + crude extract AEBxbR. (**C**) a comparison of the wound size between hydrogel + crude extract AEBxbR treatment and KitosCell^®^ treatment.

**Figure 6 pharmaceuticals-18-00752-f006:**
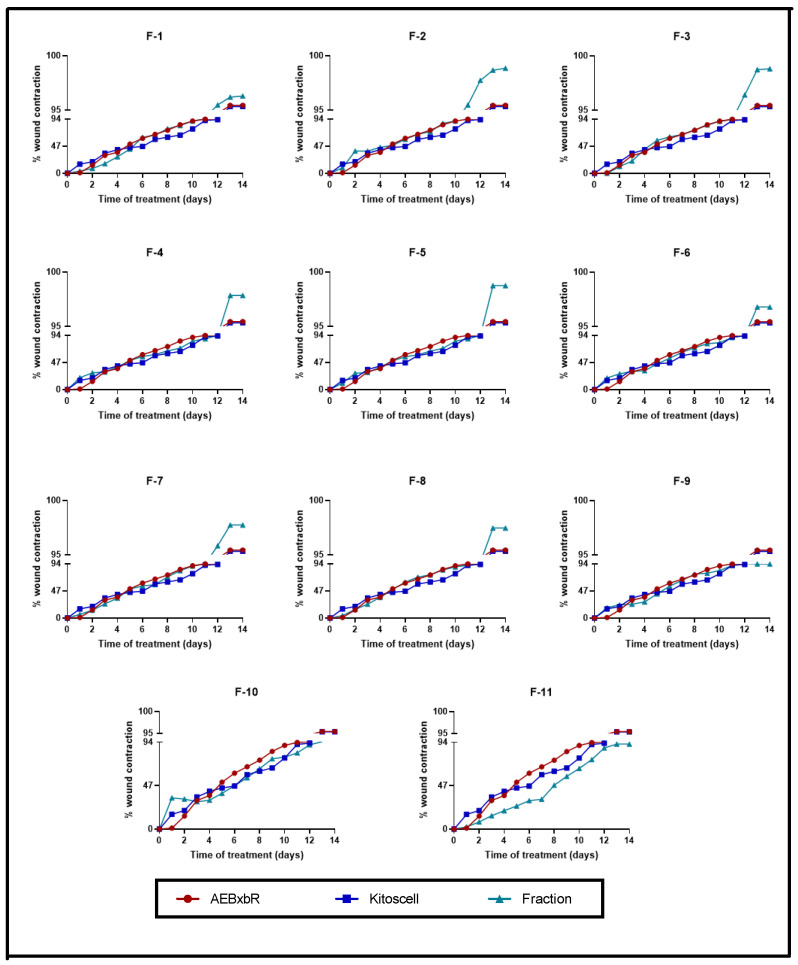
Wound contraction percentages from days 0 to 14. Values correspond to the mean (n = 6, +SEM) of the two-way ANOVA statistical analysis using Tukey’s post hoc test.

**Figure 7 pharmaceuticals-18-00752-f007:**
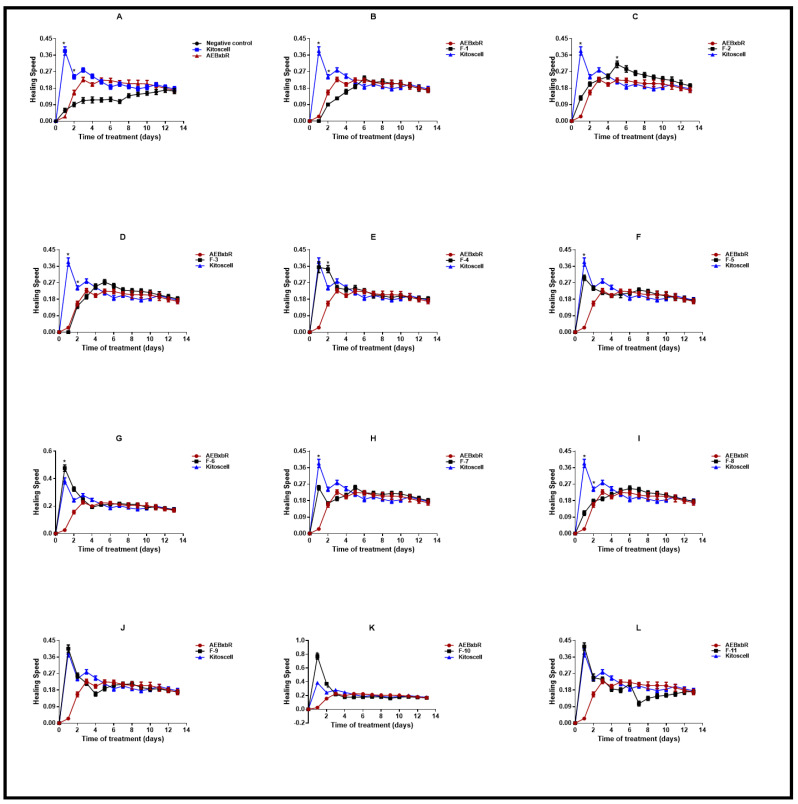
Healing speed. Values correspond to the mean (n = 6, +SEM) of the two-way ANOVA statistical analysis using Tukey’s post hoc test (* *p* < 0.001). (**A**) Healing speed of the groups of animals treated with AEBxbR. (**B**) Healing speed of the groups of animals treated with F-1. (**C**) Healing speed of the groups of animals treated with F-2. (**D**) Healing speed of the groups of animals treated with F-3. (**E**) Healing speed of the groups of animals treated with F-4. (**F**) Healing speed of the groups of animals treated with F-5. (**G**) Healing speed of the groups of animals treated with F-6. (**H**) Healing speed of the groups of animals treated with F-7. (**I**) Healing speed of the groups of animals treated with F-8. (**J**) Healing speed of the groups of animals treated with F-9. (**K**) Healing speed of the groups of animals treated with F-10. (**L**) Healing speed of the groups of animals treated with F-11.

**Figure 8 pharmaceuticals-18-00752-f008:**
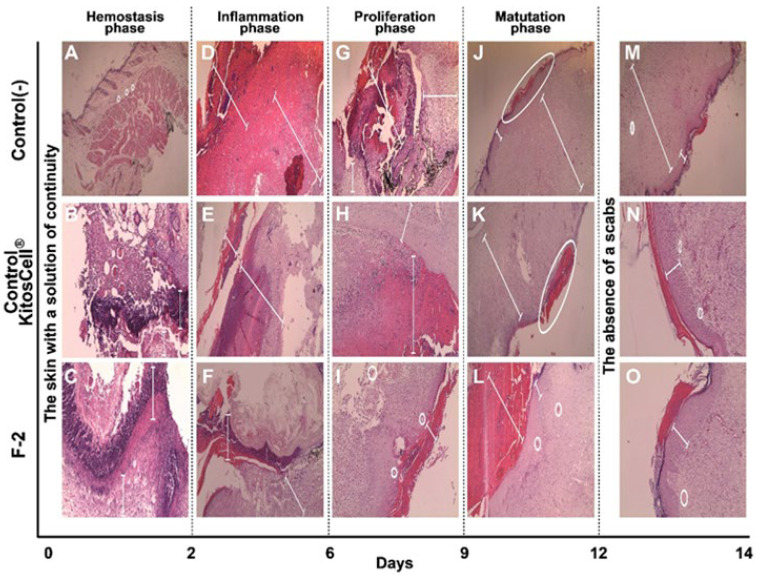
Photomicrographs of the histological sections. The wound area of skin tissue at different days of treatment, stained with hematoxylin and eosin (H&E). (**A**,**D**,**G**,**J**,**M**) correspond to the negative control treatment; (**B**,**E**,**H**,**K**,**N**) are for the positive control (KitosCell^®^); (**C**,**F**,**I**,**L**,**O**) for the F-2 treatment. Magnification: 10×.

**Figure 9 pharmaceuticals-18-00752-f009:**
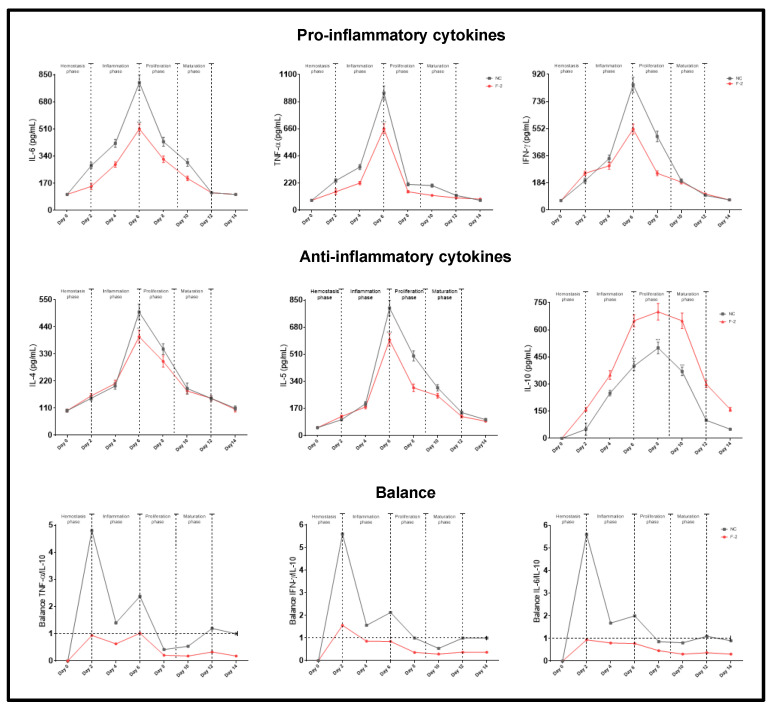
Production of pro- and anti-inflammatory cytokines. Animal groups were separated at random and treated with F-2 and/or negative control for different times, and the serum obtained was assayed as described in the Materials and Methods section for cytokine determination using the ELISA method. IL-6/IL-10, TNF-α/IL-10, and IFN-γ/IL-10 ratios. Values correspond to the mean (n = 5, +SEM) of the two-way ANOVA with Tukey’s post hoc test.

**Table 1 pharmaceuticals-18-00752-t001:** Weight and percentage yield.

Fraction Group	Weight (mg)	Yield (%)
F-1	186.6	6.22
F-2	233.8	7.79
F-3	197.9	6.60
F-4	192.3	6.41
F-5	244.0	8.13
F-6	207.2	6.91
F-7	206.0	6.87
F-8	256.8	8.56
F-9	219.8	7.33
F-10	222.2	7.41
F-11	225.7	7.52
Total	2392.3	79.74

**Table 2 pharmaceuticals-18-00752-t002:** Chromatography summary.

Eluent System	Fraction	R_f_ Values	Color	Area (cm^2^)
Water– acetonitrile (10:90)	Rutin	0.95	Red	0.15 ± 0.02
	F-1	-	-	-
	F-2	0.79; 0.91	Red	0.20 ± 0.03
	F-3	0.88	Green	-
	F-4	0.80; 0.88	Yellow	-
Water– acetonitrile (70:30)	Rutin	0.32	Orange	0.20 ± 0.03
	F-5	0.20; 0.25; 0.58	Green; Orange; Blue	-
	F-6	0.19; 0.25; 0.31	Red; Green; Orange	0.10 ± 0.03
	F-7	0.20; 0.25; 0.30	Red, Green, Orange	0.12 ± 0.03
	F-8	0.21; 0.26; 0.33; 0.38	Green; Orange	0.33 ± 0.02
	F-9	0.23; 0.27	Green; Orange	-
	F-10	0.10; 0.20; 0.25; 0.33; 0.39	Gray; Green; Orange	0.33 ± 0.02
	F-11	-	-	-
	AEBxbR	0.21; 0.27; 0.33; 0.40; 0.59	Red; Gray; Green; Orange; Blue	0.10 ± 0.04

**Table 3 pharmaceuticals-18-00752-t003:** Parameters of histologic assessment of wound.

Semi-Quantitative Method	Negative Control	Positive Control KitosCell^®^	F-2
**Day 2**			
Inflammatory infiltrate	+++	++	++
**Day 6**			
Layer granulation	+++	++	++
Ulceration	+++	++	+
Angiogenesis	-	-	++
**Day 9**			
Chronic inflammation	+++	++	+
Scab	++	+	+
Collagen fibers	-	++	+++
Fibrosis	-	-	++
**Day 12**			
Fibrosis	+++	-	+
Scab	++	++	+
Re-epithelialization, partial	-	+	++
**Day 14**			
Re-epithelialization, total	+	+	+++
Epithelium retraction	+	-	-
Fibrosis	+++	++	++

Absence: -; presence: mild +; moderate ++; marked +++.

**Table 4 pharmaceuticals-18-00752-t004:** The composition of hydrogel used as vehicle.

Compound	Amount
Propilenglycol	1 mL
Trietanolamine	0.15 mL
Water	10 mL
AEBxbR	1 mg/kg
Fractions	1 mg/kg

**Table 5 pharmaceuticals-18-00752-t005:** Experimental design for the topical evaluation of the different treatments.

Experimental Group	Treatment	
1	Negative control	Without treatment
2	Positive control	KitosCell^®^
3	Vehicle	Hydrogel
4	Hydrogel with	AEBxbR crude extract (1 mg/kg)
5	Hydrogel with	Fractions 1 up to 11 (1 mg/kg)

## Data Availability

Data is contained within the article.

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
