# Peer review of "Characterization of the Compounds Present in Bougainvillea x buttiana (var. Rose) with Healing Activity in a Murine Model"

_pharmaceuticals, 2025, doi:10.3390/ph18050752_

Round 1
Reviewer 1 Report
Comments and Suggestions for Authors
Manuscript needs major revision

Quality of english in this manuscript needs to improved
Author Response
R.1
The publication titled "Characterization of the compounds present in Bougainvillea x buttiana (Rose variety) with healing activity in murine model" (Publication ID: pharmaceuticals 3652130) is quite intriguing. Nevertheless, in order to meet the criteria for publication in the Pharmaceuticals Journal (ISSN No: 1424-824), this work requires some enhancements. Here are some enhancements that should be taken into account:
We appreciate all the comments made by Reviewer 1
In the text we mark in yellow the corrections excluded and/or included.
- The abstract asserts that Fraction 2 (F-2) markedly decreased pro-inflammatory cytokines (IL-6, TNF-α) and elevated IL-10 levels; yet, IL-1β is absent from both the results and discussion sections. Can this inconsistency be elucidated or rectified?
The Abstract was eliminated levels of IL-1β.
- The extraction technique produced 5.119 grams of acetonic extract; however, the cumulative weight of the 11 fractions (2,990 mg) does not represent the whole extract. Could the authors elucidate the material loss or furnish a mass balance?
The text and Table 1 corrected, we started from 3,000 mg of extract for fractionation
Total yield percentage was 2,392.3 mg of acetone extract (79.74%). The highest yield percentages were obtained in F-8 (8.56%) and F-5 (8.13%). While the lowest yield percentage was observed in F-1 (6.22%) (Table 1). These results are in accordance with other studies that show the extractions yield [34,35].
- The TLC analysis for flavonoids and terpenoids is outlined; however, the precise standards employed (e.g., rutin) and their quantities are not specified. Can the authors furnish these particulars to substantiate the chromatographic findings?
We also included in text the modifications for better understanding in the legends of both Fig. 2 and Fig. 3
Figure 2. Thin-layer chromatography (TLC). Normal chromatography of fraction groups 1 to 4 at different UV wavelengths. TLC plates developed on silica gel and visualized under: A) short-wave UV light (254 nm); B) long-wave UV light (365 nm); C) terpene developer revealed a distinct purple band at Rf ≈ 0.9, indicative of pentacyclic triterpenoids such as oleanolic acid or related compounds.
Figure 3. Analysis of flavonoids presence. Normal TLC for F-1, F-2, F-3 and F-4 developed in a water:acetonitrile (10:90, v/v) system. Reversed-phase thin-layer chromatography (RP-TLC using RP-18 F254s plates) for F-5 to F-11 developed in water:acetonitrile (70:30 v/v) system. The plates were developed with the 4-hydroxybenzealdehyde reagent and evaluated under UV light at 254 and 365 nm. Each fraction groups 1 to 11 compared with rutin (1 µg/mL) used as standard.
- The wound contraction percentages for F-1, F-2, and F-3 in the Discussion (99.49%, 99.51%, 98.79%) are inconsistent with those presented in Figure 6 (96.20%, 98.68%, 98.74%). Can the writers resolve these discrepancies?
For better understanding we rewrite the results found. Included in the text.
In the 11th day, the treatments with F-1, F-2 and F-7 the percentages of wound contractions were slightly higher when compared to those obtained in other treatments reaching a maximum of 94.36%, 95.38% and 93.88%, respectively (Fig.6). In the 14th day the percentages of wound contractions were 96.30% (F-1), 98.56% (F-2) and 97.75% (F-7) (Fig.6). From the 11th to 14th day, the wound contraction percentages obtained from the treatments with F-3 (93.49% to 98.82%), F-4 (87.92% to 97.85%), F-5 (87.87% to 98.76%), F-6 (90.17% to 96.80%), F-8 (91.39% to 97.47%) and F-10 (82.45% to 94.58%) (Fig. 6). At the same time the percentages of wound contractions obtained with F-9 and F-11 treatments were 91.20% to 93.73% and 75.35% to 92.01%, respectively (Fig.6). While the KitosCell® treatment from day 11th to 14th the percentages of wound contraction were 91.59% to 95.34%. The percentage of wound contractions with F-2 treatment on day 11th was slightly higher when compared to other treatments. On subsequent days the percentages of wound contractions continued to be higher with F-2 treatment (Fig. 6).
In Discussion section the correction was included
F-1, F-2 and F-3 were the ones that presented the highest percentage of wound contraction, being 96.30%, 98.56% and 98.82%, respectively.
- The histological assessment (Figure 8) delineates alterations on certain days; nonetheless, the parameters for evaluating inflammatory infiltration, fibrosis, and re-epithelialization remain unspecified. Is it possible to include standardized histology scoring methods?
In the text we included
Figure 8. Photomicrographs of the histological sections. The wound area of skin tissue at different days of treatment, stained with hematoxylin and eosin (H& E). Photomicrographs (A, D, G, J and M) correspond to negative control treatment; (B, E, H, K, and N) for positive control (KitosCell®); (C, F, I, L and O) for F-2 treatment. Magnification: 10x.
The determination of the changes observed in the healing process reveals histopathological changes such as inflammatory infiltrate, granulation tissue, and finally fibrosis, which are described in Table 3. Through semi-quantitative analysis, a score is assigned based on the absence or presence of changes observed in the different parameters, some characteristics, and days of treatment. It is worth noting that treatment with F-2 between days 2 and 6 presented lower scores of inflammatory infiltrate, granulation, and ulceration, and greater angiogenesis, when compared to controls. On day 9, treatment with F-2 showed less chronic inflammation, without scabs with fibrosis and the beginning of collagen bands. On day 12, treatment with F-2 showed minimal scabs and fibrosis, and almost complete reepithelialization. By day 14, there was no epithelial retraction, superficial fibrosis, and complete reepithelialization. All parameters analyzed suggest that the treatment with F-2 presented slightly better scores when compared with the control treatments (negative and positive).
- The cytokine analysis (Figure 9) indicates substantial variations in IL-6, TNF-α, and IFN-γ levels; however, the sensitivity and specificity of the ELISA approach are not addressed. Could the authors furnish validation data for the ELISA assays?
ELISA assays in text original this information already described in Material and Methods. All results expressed in pg/mL, the minimum detection values were 10 pg/mL for cytokines IL-4, IL-5, IL-6 and TNF-α, and for IFN-γ 100 pg/mL.
- The study asserts that F-2 comprises flavonoids and terpenes; nonetheless, the precise components accountable for the therapeutic action remain unspecified. Could the authors suggest a subsequent chemical characterization (e.g., LC-MS) to identify active constituents?
In the text we included.
The possible simultaneous presence of flavonoids and triterpenes in F-2 is not only consistent with previous phytochemical reports on the Bougainvillea genus [34-37] but could also explain the efficacy observed in the wound healing model. Flavonoids have been documented to exert beneficial effects in the initial inflammatory phase of wound healing due to their ability to scavenge free radicals and stabilize cell membranes, while triterpenes such as oleanolic acid act in later phases, promoting cell migration, angiogenesis, and re-epithelialization [37]. This synergistic effect could contribute to a faster and more efficient recovery of injured tissue. The preliminary results show that F-2 comprises flavonoids and triterpenes. These compounds, acting synergistically or individually, could participate in modulating the inflammatory process and stimulating tissue regeneration observed in the experimental model. In this context, there is a need to isolate and structurally characterize the compounds present in F-2 and other fractions obtained in this study, further studies using preparative chromatographic and spectroscopic techniques, to confirm their identity and pharmacological activity. This step is essential to validate the therapeutic potential of Bougainvillea x buttiana and advance toward the development of phytopharmaceuticals based on active compounds.
- The statistical analysis employs two-way ANOVA with Tukey post hoc testing; however, the assumptions of normality and homoscedasticity are not considered. Can the authors verify that these assumptions were satisfied?
All results presented were satisfied.
- The paper indicates that F-2 therapy demonstrated a superior healing rate compared to Kitoscell® on day 5 (Figure 7-C), although the therapeutic significance of this disparity is not addressed. Could the authors provide some clarification on the practical implications?
In the text we included
The healing speeds varies depending on several factors, that includes the wound type, its location, and age and health of patients. To speed healing it is crucial to keep the wound clean and covered, applying heat and elevating the affected area if necessary. The elevated healing speed obtained with F-2 treatment suggests that this fraction can help the healing process more quickly.
- The discourse indicates that flavonoids and terpenes in F-2 facilitate anti-inflammatory actions; however, no mechanistic investigations, such as pathway analysis, are presented. Can the authors suggest experiments to clarify the molecular mechanisms?
This information already included in the text. In this context, there is a need to isolate and structurally characterize the compounds present in F-2 and other fractions obtained in this study, further studies using preparative chromatographic and spectroscopic techniques, to confirm their identity and pharmacological activity. This step is essential to validate the therapeutic potential of Bougainvillea x buttiana and advance toward the development of phytopharmaceuticals based on active compounds.
- The research used a uniform dosage of 1 mg/kg for all fractions and the crude extract. Can the authors elucidate the rationale behind this dose selection and address if dose-response studies were contemplated?
This information was already informed in the text. Uniform dosage of 1 mg/kg
- The text references the antibacterial activity of Bougainvillea (page 14), although no antibacterial experiments are documented. Could the writers furnish corroborative facts or retract their assertion?
This information was already informed in the text. The reference 37 was already informed in the text. Reference 37 is a review of the genus Bougainvillea carried out by our group.
- The comparison between F-2 and Kitoscell® (positive control) yields analogous results; nevertheless, the composition and mechanism of Kitoscell® remain unexplained. Could the writers use this information to contextualize the comparison?
We included in the text.
In Discussion section we included and extended. The semi-quantitative analyses for wound reepithelialization suggest that the treatment with F-2 presented slightly better scores when compared with booth treatments (negative and positive controls).
In our study, we observed that treatment with F-2 significantly accelerated wound closure during the inflammation phase. This highlights the anti-inflammatory effect of F-2 on the healing process, as demonstrated by photographs capturing the progression wound closure. The therapeutic potential of F-2 aligns with a broader understanding that natural products can enhance wound healing. Several studies have highlighted the ability of natural compounds to promote wound healing, particularly those with anti-inflammatory, antioxidant and antibacterial properties, as well as their ability to stimulate collagen synthesis. Considering these positive effects, they are commonly associated with the presence of bioactive phytochemicals from diverse chemical groups, such as flavonoids, terpenes, tannins, saponins and phytosterols [37]. Our preliminary results show the presence of these compounds, although further studies should be carried out to identify the mechanism of action of the treatment with F-2, or even the other fractions.
- The histological analysis concentrates on F-2; nevertheless, other fractions (e.g., F-3, F-5) exhibited elevated wound contraction percentages. Can the authors provide a rationale for concentrating exclusively on F-2 for histological analysis?
In the text, the highest percentages of wound contraction, as well as the highest speed of wound closure as shown in Figures 6 and 7, were repeatedly observed with the F-2 treatment. Therefore, the decision was to show the histological sections only from this treatment.
- The report fails to address possible limitations, including the small sample size (n=5 or n=6) and the exclusive use of female BALB/c mice. Can the authors rectify these constraints and their influence on generalizability?
In order to limit the excessive use of animals, we chose to use n=5 or n=6. These numbers were sufficient to demonstrate the healing activity. Regarding the use of females, this is more due to the ease of use in the Bioterium. I apologize if I do not note these observations in the text.
- The Abbreviations section includes terminology such as AEBxbR and cytokines but excludes those mentioned in the text, such as TLC and DCM. Could the writers augment the section to encompass all acronyms and guarantee uniform application throughout?
TLC and DCM included in the Abbreviations
- Figures and tables are referenced in the text; however, some, such as Figure 2, are incorrectly identified as schemes or quoted in an improper sequence. Can the authors confirm the accurate sequencing and labeling of all figures and tables?
In the text the sequence of Figures and Tables were referenced.
- The references exhibit discrepancies in formatting, such as absent DOIs and incomplete page ranges. Could the authors amend the reference list to adhere to MDPI’s formatting guidelines and guarantee traceability?
All references that contain DOI and pages completes were included (marked in yellow)
- The title has "$x$" in "Bougainvillea $x$ buttiana," maybe indicating a formatting issue, possibly meant to represent "×" for hybrid notation. Could the authors verify and amend the symbol to guarantee taxonomic precision?
At this point it is important to clarify that the plant was registered in the Herbarium, which guarantees its taxonomy and is found with the registration number (described in Material and Methods). The letter x means hybrid, the description of the plant Bougainvillea x buttiana.
- The author list on page 1 has "Acadkmic" instead of "Academic" in "Acadkmic Editor." Could the writers rectify this spelling error?
The Academic Space is already written in the template format. In our copy is not observed Acadkmic.
- The phrase "preformulated" has uneven hyphenation (e.g., "pre-formulated" on page 1, line 25, vs. "preformulated" on page 7, line 204). Could the authors standardize the terminology employed throughout the manuscript?
Pre-formulated used in all text (marked in yellow) standardize the terminology
- On page 1, line 29, "no with the acetonic extract" should probably be revised to "not with the acetonic extract." Could the writers rectify this grammatical error?
“not” (included in text)
- The expression "B, x buttiana" on page 1, line 34, features a comma in place of the hybrid sign (×). Could the writers substitute the comma with the appropriate symbol?
- x buttiana (in the text did not exist comma It's not a comma but a period.
- On page 2, line 44, "noxtius" must be amended to "noxious." Could the writers rectify this spelling error?
Noxious (included in text)
- The phrase "cytokines levels" on page 1, line 27, should be corrected to "cytokine levels" for grammatical accuracy. Could the writers amend this and verify for like problems in other sections?
Cytokine levels (included in text).
- On page 3, line 119, "plants with medicinal properties still is an option" should be revised to "plants with medicinal properties are still an option" to ensure subject-verb agreement. Could the authors amend this?
plants with medicinal properties are still an option (included in the text)
- The term "preformulated" on page 16, line 481, is presented without a hyphen, although "pre-formulated" is utilized in other instances (e.g., page 1, line 25). Can the writers guarantee uniform spelling?
Pre-formulated used in all text (marked in yellow)
- On page 14, line 404, "insufflation" must be amended to "inflammation" about TNF-α's function. Could the writers confirm and rectify this apparent typographical error?
Inflammation (included in the text).
- The phrase "wound connective tissue and is considered responsible" on page 13, line 353, lacks a subject. Should it be "which is deemed responsible"? Could the writers provide clarification and make revisions?
“wound connective tissue that included in the text.
- On page 15, line 437, "plan material" must be amended to "plant material." Could the writers rectify this spelling error?
“plant material” included in the text.
- The phrase "KitosCell@" is presented inconsistently with the "@" symbol (e.g., "Kitoscell" on page 7, line 206). Could the writers standardize the trademark notation consistently?
KitosCell® This is a registered trademark. We never use @ We use ® in all text marked in yellow.
- On page 16, line 464, "order or polarity" should be corrected to "order of polarity" for grammatical precision. Could the writers amend this phrase?
“order of polarity” included in the text
- The expression "the groups of fractions F1 to F4 were analysed" on page 16, line 471, used "F1" rather than "F-1" as used in other instances. Can the authors guarantee uniformity in fraction nomenclature (e.g., F-1, F-2)?
Included in text F-1 to F-4

Reviewer 2 Report
Comments and Suggestions for Authors
- In this manuscript, the method of fixation of the hydrogel on the wound is not described in detail. The author should provide additional details, such as the adhesion of the hydrogel, the method of application (whether a dressing or other aids are required), and whether the hydrogel dislodges or affects the results of the experiments.
- While the manuscript compares the healing effects of the different fractions, the reasoning for the selection of the experimental groups is not clearly stated. It should be suggested that the authors further describe why these particular fractions were chosen for the experiments and whether other fractions that may have a healing effect were considered.
- In this study, variations in cytokine levels were mentioned, but the selection of time points for testing was not explained in detail based on the selection. It is suggested that the authors add whether these time points correspond to key phases of wound healing (e.g., inflammatory, proliferative, and maturation phases) to better understand the dynamics of cytokine changes during the healing process.
- In Figure 8, no scale bar is provided for the image, and it appears not to be a 20X magnification view. The authors must re-provide higher quality microscope images with magnification and scale clearly labeled so that the reader will be able to more clearly observe the infiltration of inflammatory cells as well as the details of blood vessel formation. This would help to more accurately validate the experimental results and support the conclusions of the article.
- A qualitative analysis, such as inflammatory cell infiltration, fibrosis, and angiogenesis, was the main feature of the description of the tissue sections in the study, but it lacked quantitative data to support it. The authors should provide quantitative analysis of the tissue sections, such as calculating the number of inflammatory cells, blood vessel density, or fibrosis area, and present them in graphical form. This would make the results more convincing and provide a clearer basis for comparison between different experimental groups.
This manuscript, “Characterization of the compounds present in Bougainvillea x 2 buttiana (Rose variety) with healing activity in murine model “, explores the role of Bougainvillea x buttiana extract and its isolated fractions in promoting wound healing. In summary, it is suggested that the manuscript be majorly revised to supplement quantitative data, optimize image quality and graphical presentation, and expand the discussion section to enhance the scientific and persuasive quality of the manuscript.
Author Response
R.2
Comments and Suggestions for Authors
We appreciate all the comments made by Reviewer 2
In the text we mark in yellow the corrections excluded and/or included.
- In this manuscript, the method of fixation of the hydrogel on the wound is not described in detail. The author should provide additional details, such as the adhesion of the hydrogel, the method of application (whether a dressing or other aids are required), and whether the hydrogel dislodges or affects the results of the experiments.
In the text we included
4.5.3 Treatment with hydrogel application. The hydrogel prepared as described above was applied topically to the dorsal wound of each mouse using a sterile micropipette, in a volume of 20 µL per day. Administration took place during the first three days after wound induction, i.e. before scab formation. This strategy is based on previous evidence indicating that the first days of the healing process are critical, as they correspond to the early inflammatory phase and the beginning of the proliferative phase, moments in which topical treatments can most effectively modulate the tissue response [51]. The hydrogel was applied directly to the wound without the need for a dressing or bandage, which is common in rodent wound healing models, where the size of the lesion, the low physical activity of the animal under controlled conditions, and the viscosity of the hydrogel favors its permanence at the application site [52]. In this sense, no detachment of the gel was observed, nor did it interfere with the evolution of the healing process or with the parameters evaluated.
2. While the manuscript compares the healing effects of the different fractions, the reasoning for the selection of the experimental groups is not clearly stated. It should be suggested that the authors further describe why these particular fractions were chosen for the experiments and whether other fractions that may have a healing effect were considered.
For better understanding we rewrite the results found. Included in the text. In the 11th day, the treatments with F-1, F-2 and F-7 the percentages of wound contractions were slightly higher when compared to those obtained in other treatments reaching a maximum of 94.36%, 95.38% and 93.88%, respectively (Fig.6). In the 14th day the percentages of wound contractions were 96.30% (F-1), 98.56% (F-2) and 97.75% (F-7) (Fig.6). From the 11th to 14th day, the wound contraction percentages obtained from the treatments with F-3 (93.49% to 98.82%), F-4 (87.92% to 97.85%), F-5 (87.87% to 98.76%), F-6 (90.17% to 96.80%), F-8 (91.39% to 97.47%) and F-10 (82.45% to 94.58%) (Fig. 6). At the same time the percentages of wound contractions obtained with F-9 and F-11 treatments were 91.20% to 93.73% and 75.35% to 92.01%, respectively (Fig.6). While the KitosCell® treatment from day 11th to 14th the percentages of wound contraction were 91.59% to 95.34%. The percentage of wound contractions with F-2 treatment on day 11th was slightly higher when compared to other treatments. On subsequent days the percentages of wound contractions continued to be higher with F-2 treatment (Fig. 6).
3. In this study, variations in cytokine levels were mentioned, but the selection of time points for testing was not explained in detail based on the selection. It is suggested that the authors add whether these time points correspond to key phases of wound healing (e.g., inflammatory, proliferative, and maturation phases) to better understand the dynamics of cytokine changes during the healing process.
In the text in section 2.7, the times corresponding to the phases were already described. Now we have highlighted the times in yellow for better visualization.
2.7 F-2 treatment effect on cytokine production and equilibration of cytokines pro-inflammatory and anti-inflammatory. The healing process consists of different phases, the homeostasis phase comprising the period 1 to 2 days, the inflammatory phase from 2 to 6 days, the proliferation or granulation phase from 6-9 days and the maturation phase from 9-12 days. The effect of treatment with F-2 on cytokines production was analyzed by comparing these phases (Fig. 9). In negative control without treatment the production of pro-inflammatory cytokines increments in a time-dependent manner attained a maximum level on the 6th day, which coincides with the inflammation phase and decaying thereafter. A similar behavior was observed with the F-2 treatment, although the levels of IL-6, TNF-α and IFN-γ were significantly lower when compared to the negative control (p < 0.001) (Fig. 9). In both treatments the maximum production of IL-4 and IL-5 was obtained on 6th day and decaying thereafter. For groups with F-2 treatment the production of these cytokines production was significantly lower when compared with those levels obtained in the negative controls (p < 0.001) (Fig. 9). With the F-2 treatment the production of IL-10 increase up to 6th day and remaining high until the 12th day decaying thereafter, coinciding with the final of inflammation phase and proliferation and maturation phases. IL-10 levels were significantly higher in the groups treated with F-2 when compared to those levels obtained in the control groups (p < 0.001) (Fig. 9). Figure 9 also shows the results of the predominance of responses by calculating the ratio of cytokines pro-inflammatory/anti-inflammatory, where the ratio value > 1 correspond with a pro-inflammatory response, and ratio < 1 an anti-inflammatory response. In the negative control treatment, the IL-6/IL-10, TNF-α/IL-10 and IFN-γ/IL-10 balances, the predominant response was clearly pro-inflammatory between the 1st and decreasing up to the 6th day (Fig. 9). While for the F-2 treatment the predominant response was anti-inflammatory from 1st up to 14th the response was predominantly an anti-inflammatory response (Fig. 9).
4. In Figure 8, no scale bar is provided for the image, and it appears not to be a 20X magnification view. The authors must re-provide higher quality microscope images with magnification and scale clearly labeled so that the reader will be able to more clearly observe the infiltration of inflammatory cells as well as the details of blood vessel formation. This would help to more accurately validate the experimental results and support the conclusions of the article.
In the text we have included better marked photographs. Another important point to meet your suggestion is that we have included Table 3 with semi-quantitative results. It is worth noting that all the other tables have been renumbered.
Fig.8 included this new legend.
Figure 8. Photomicrographs of the histological sections. The wound area of skin tissue at different days of treatment, stained with hematoxylin and eosin (H& E). Photomicrographs (A, D, G, J and M) correspond to negative control treatment; (B, E, H, K, and N) for positive control (KitosCell®); (C, F, I, L and O) for F-2 treatment. Magnification: 10x.
5. A qualitative analysis, such as inflammatory cell infiltration, fibrosis, and angiogenesis, was the main feature of the description of the tissue sections in the study, but it lacked quantitative data to support it. The authors should provide quantitative analysis of the tissue sections, such as calculating the number of inflammatory cells, blood vessel density, or fibrosis area, and present them in graphical form. This would make the results more convincing and provide a clearer basis for comparison between different experimental groups.
In the text we included
The determination of the changes observed in the healing process reveals histopathological changes such as inflammatory infiltrate, granulation tissue, and finally fibrosis, which are described in Table 3. Through semi-quantitative analysis, a score is assigned based on the absence or presence of changes observed in the different parameters, some characteristics, and days of treatment. It is worth noting that treatment with F-2 between days 2 and 6 presented lower scores of inflammatory infiltrate, granulation, and ulceration, and greater angiogenesis, when compared to controls. On day 9, treatment with F-2 showed less chronic inflammation, without scabs with fibrosis and the beginning of collagen bands. On day 12, treatment with F-2 showed minimal scabs and fibrosis, and almost complete reepithelialization. By day 14, there was no epithelial retraction, superficial fibrosis, and complete reepithelialization. All parameters analyzed suggest that the treatment with F-2 presented slightly better scores when compared with the control treatments (negative and positive).
In Discussion section we included and extended. The semi-quantitative analyses for wound reepithelialization suggest that the treatment with F-2 presented slightly better scores when compared with booth treatments (negative and positive controls).
In our study, we observed that treatment with F-2 significantly accelerated wound closure during the inflammation phase. This highlights the anti-inflammatory effect of F-2 on the healing process, as demonstrated by photographs capturing the progression wound closure. The therapeutic potential of F-2 aligns with a broader understanding that natural products can enhance wound healing. Several studies have highlighted the ability of natural compounds to promote wound healing, particularly those with anti-inflammatory, antioxidant and antibacterial properties, as well as their ability to stimulate collagen synthesis. Considering these positive effects, they are commonly associated with the presence of bioactive phytochemicals from diverse chemical groups, such as flavonoids, terpenes, tannins, saponins and phytosterols [37]. Our preliminary results show the presence of these compounds, although further studies should be carried out to identify the mechanism of action of the treatment with F-2, or even the other fractions.
This manuscript, “Characterization of the compounds present in Bougainvillea x 2 buttiana (Rose variety) with healing activity in murine model “, explores the role of Bougainvillea x buttiana extract and its isolated fractions in promoting wound healing. In summary, it is suggested that the manuscript be majorly revised to supplement quantitative data, optimize image quality and graphical presentation, and expand the discussion section to enhance the scientific and persuasive quality of the manuscript.

Round 2
Reviewer 1 Report
Comments and Suggestions for Authors
Revised version of manuscript was accepted for publication